# Targeted next-generation sequencing has incremental value in the diagnostic work-up of patients with suspect pancreatic masses; a multi-center prospective cross sectional study

**Friso B. Achterberg**[1☯], **Babs G. Sibinga Mulder**[1☯], **Quisette P. Janssen**[2], **Bas Groot Koerkamp**[2], **Lieke Hol**[3], **Rutger Quispel**[4], **Bert A. Bonsing**[1], **Alexander L. Vahrmeijer**[1], **Casper H. J. van Eijck**[2], **Daphne Roos**[5], **Lars E. Perk**[6], **Erwin van der Harst**[7], **Peter-Paul L. O. Coene**[7], **Michail Doukas**[8], **Frank M. M. Smedts**[9], **Mike Kliffen**[10], **Marie-Louise F. van Velthuysen**[8], **Valeska Terpstra**[11], **Arantza Farina Sarasqueta**[12], **Hans Morreau**[13], **J. Sven D. Mieog**[1]*

1 Department of Surgery, Leiden University Medical Center, Leiden, The Netherlands, 2 Department of Surgery, Erasmus MC, University Medical Center, Rotterdam, The Netherlands, 3 Department of Gastro-Enterology, Maasstad Hospital, Rotterdam, The Netherlands, 4 Department of Gastro-Enterology, Reinier de Graaf Gasthuis, Delft, The Netherlands, 5 Department of Surgery, Reinier de Graaf Gasthuis, Delft, The Netherlands, 6 Department of Gastro-Enterology, Haaglanden Medical Center, The Hague, The Netherlands, 7 Department of Surgery, Maasstad Hospital, Rotterdam, The Netherlands, 8 Department of Pathology, Erasmus Medical MC, University Medical Center, Rotterdam, The Netherlands, 9 Department of Pathology, Reinier de Graaf Gasthuis, Delft, The Netherlands, 10 Department of Pathology, Maasstad Hospital, Rotterdam, The Netherlands, 11 Department of Pathology, Haaglanden Medical Center, The Hague, The Netherlands, 12 Department of Pathology, Amsterdam University Medical Centers, Amsterdam, The Netherlands, 13 Department of Pathology, Leiden University Medical Center, Leiden, The Netherlands

☯ These authors contributed equally to this work.
* j.s.d.mieog@lumc.nl

**Data Availability Statement:** The data of the study will not be directly available on an open-source or

## Abstract

### Background

The diagnostic process of patients with suspect pancreatic lesions is often lengthy and prone to repeated diagnostic procedures due to inconclusive results. Targeted Next-Generation Sequencing (NGS) performed on cytological material obtained with fine needle aspiration (FNA) or biliary duct brushing can speed up this process. Here, we study the incremental value of NGS for establishing the correct diagnosis, and subsequent treatment plan in patients with inconclusive diagnosis after regular diagnostic work-up for suspect pancreatic lesions.

### Methods

In this prospective cross-sectional cohort study, patients were screened for inclusion in four hospitals. NGS was performed with AmpliSeq Cancer Hotspot Panel v2 and v4b in patients with inconclusive cytology results or with an uncertain diagnosis. Diagnostic results were evaluated by the oncology pancreatic multidisciplinary team. The added value of NGS was determined by comparing diagnosis (malignancy, cystic lesion or benign condition) and proposed treatment plan (exploration/resection, neoadjuvant chemotherapy, follow-up,

public library. The study data contains personal (patient) data which cannot be anonymized without impeding data quality. Anonymizing the data will impair the possibility to reproduce the study data. Along with the manuscript, a variable library is submitted, which contains the variables used to produce the manuscript data in conclusion. Data are available upon request from the Leiden University Medical Center Institutional Ethics Committee. A research proposal can be submitted to the corresponding author (J.S.D. Mieog, MD PhD) or the pancreatic clinical care and research center (pancreas@lumc.nl) along with the required variables. This proposal shall be reviewed by the scientific team and ethical committee of the institute. In case of a positive review the inquirer is asked to submit a data transfer agreement.

**Funding:** The manuscript was funded by a grant of the Dutch Cancer Society (KWF) granted to the corresponding authors J.S.D. Mieog. The funders had no role in study design, data collection and analysis, decision to publish, or preparation of the manuscript.

**Competing interests:** The authors have declared that there are no competing interests.

palliation or repeated FNA) before and after integration of NGS results. Final histopathological analysis or a 6-month follow-up period were used as the reference standard in case of surgical intervention or non-invasive treatment, respectively.

## Results

In 50 of the 53 included patients, cytology material was sufficient for NGS analysis. Diagnosis before and after integration of NGS results differed in 24% of the patients. The treatment plan was changed in 32% and the diagnosis was substantiated by the NGS data in 44%. Repetition of FNA/brushing was prevented in 14% of patients. All changes in treatment plan were correctly made after integration of NGS. Integration of NGS increased overall diagnostic accuracy from 68% to 94%.

## Interpretation

This study demonstrates the incremental diagnostic value of NGS in patients with an initial inconclusive diagnosis. Integration of NGS results can prevent repeated EUS/FNA, and can also rigorously change the final diagnosis and treatment plan.

## Introduction

Diagnostic work-up for patients with a suspect lesion in the pancreas can be difficult, frequently leading to inconclusive diagnoses. Current imaging techniques are limited for differentiating between pancreatic cancer, inflammation, benign lesions or preneoplastic lesions [1]. Preoperative pathological confirmation is required to establish an accurate treatment plan and prevent futile surgery. Moreover, increased use of neoadjuvant treatment in pancreatic duct adenocarcinomas (PDAC) requires adequate tissue analysis to prevent overtreatment or treatment of benign lesions. Endoscopic ultrasound-guided fine needle aspiration (EUS-FNA) and biliary duct brushing can be performed to obtain a cytopathological diagnosis. However, a significant limitation of this technique is the high false positive/negative rate or inconclusive results, which have been reported in up to a third of the cases. These are mainly caused by sampling error, suboptimal sample quality, low cellular yield and the presence of an intense desmoplastic stromal reaction [2]. As a consequence, the diagnostic process of patients with a suspect pancreatic lesion is often lengthy due to repetitive diagnostics like EUS-FNA and subject to frequent misdiagnosis potentially leading to a delay or inappropriate treatment [3].

Targeted Next-Generation Sequencing (NGS) on FNA-derived DNA can identify pathogenic variants in known driver genes of malignancies, and can therefore be used to confirm neoplasia [4]. Moreover, NGS has proven to be useful in distinguishing benign from malignant pancreatic lesions [5]. For ultra-deep sequencing using NGS panels that target hotspot gene variants such as *KRAS, TP53, BRAF and PTEN* only a limited quantity of material is required, up to 100 cells [6].

In a previous study, we evaluated 70 consecutive patients with a suspect pancreatic lesion and reported a diagnostic accuracy, sensitivity and specificity of NGS of 94%, 93% and 100%, respectively [5]. Since NGS might be superfluous in all patients with a suspect pancreatic lesion, the focus of this study was to evaluate the applicability of NGS in patients with inconclusive cytology material. Therefore, the aim of this study was to determine the incremental diagnostic value of NGS in the diagnostic process and treatment plan proposition in patients with a suspect pancreatic lesion with an inconclusive cytopathologic diagnosis.

## Materials and methods

### Study design

This multicenter study was conducted in the Leiden University Medical Center (LUMC), Erasmus Medical Center, Maasstad hospital and Reinier de Graaf hospital. The study was approved by the medical ethical reviews board of the LUMC (Dutch trial register: NTR7006).

Patients above 18 years of age and with a suspected malignancy based on a solid or cystic pancreatic or periampullary lesion (papilla or distal bile duct) on recent multi-phase abdominal CT or MRI scan were evaluated for inclusion at the first multidisciplinary tumor board (MDT) meeting (Fig 1). Available cytology samples for pathological assessment were obtained by either by EUS-FNA or common bile duct brushing as part of the standard work-up. Patients without a definitive diagnosis, or with an inconclusive cytopathologic diagnosis after the first MDT for pancreatic lesions were eligible for this study and included after written consent was acquired. Thereafter, NGS analysis was performed on the cytology sample. Prior to the second MDT meeting, the cytology sample was reviewed by a central review board of the LUMC department of Pathology by dedicated pancreatic pathologists (A.F.S., H.M.). Samples were classified based on morphological assessment as: 1) normal/atypia, 2) dysplasia or, 3) inconclusive (in case of too few cells or bloody material) [5].

Based on all available diagnostic information, a probability diagnosis was proposed at each of the two MDT meetings, either: 1) malignancy, including pancreatic ductal adenocarcinoma, distal cholangiocarcinoma, ampullary carcinoma, cystic lesion with malignant degeneration including high grade dysplasia, 2) benign condition, including (autoimmune) pancreatitis, pseudocyst, or, 3) benign cystic lesions, including mucinous cystic neoplasm and/or, low grade IPMN (not malignant degenerated). Furthermore, the most suitable treatment for the probable diagnosed condition was determined as either: 1) surgical exploration with possible resection (resection was refrained in case of a metastatic of locally advanced disease), 2) neoadjuvant therapy for study purpose or in case of locally advanced disease, 3) palliation in case of locally advanced disease, metastatic disease or patient unfit for surgery, 4) follow-up for benign conditions or 5) repetition of diagnostic FNA. For patients diagnosed with low grade IPMN/MCN, both surveillance or resection are widely accepted treatment options. The ultimate decision differs per patient and depends on the expertise of the MDT members.

The cytology samples were then analyzed with NGS, using the AmpliSeq Cancer Hotspot Panel v2 and v4b, as previously described [5]. AmpliSeq Panel v2 was replaced during the study for v4b. All regions covered in v2 were present in v4b and have the same performance. The results were reported as: 1) *"no molecular support for dysplasia"* if no pathogenic variants were identified, 2) *"a proliferative lesion with at least low grade dysplasia (LGD)"* in case of *KRAS* or *GNAS* class 4 or 5 DNA variants with sufficient coverage and high frequent variant reads on target, or the sole finding of *ATM, CDKN2A, PTEN or APC DNA variants*. The sole finding of a *KRAS* pathogenic variant with low frequency could also be suggestive for pancreatitis. 3) "*molecularly at least high grade dysplasia (HGD)"* if at least a *TP53* or *SMAD4* DNA variant was identified or 4) "*inconclusive*" if NGS was not possible or reliable on the presented material.

After integration of the NGS results during the second MDT meeting, the diagnosis and treatment plan could be changed or confirmed. Furthermore, the MDT could ignore the NGS results if too contradictive or inconclusive results or if the clinical suspicion of a malignancy was too high (i.e. false-negative NGS results). Consequently, only the availability of the NGS results differed between both MDT meetings.

The final diagnosis was determined, either based on histopathological assessment of the resected specimen or by evaluation during follow-up after at least six months. NGS analysis of

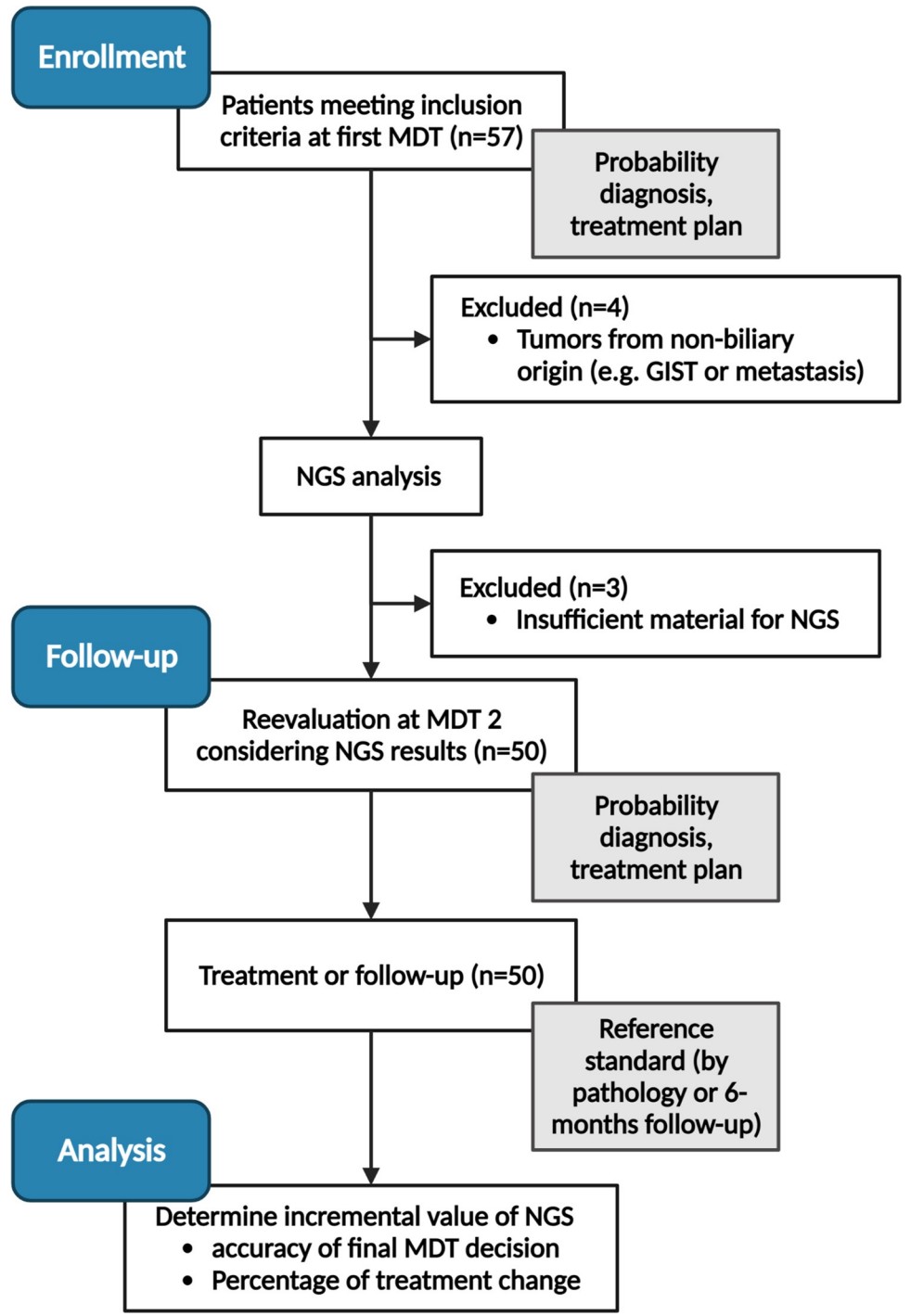

**Fig 1. CONSORT flowchart.**

the resected specimen was not repeated since correlation was already validated in an earlier study [7].

Fig 1 gives an overview of the study design.

Patients were included at the first MDT meeting after inconclusive cytology or inconclusive diagnosis. Prior to the second MDT meeting, NGS analysis was performed on the cytology

samples, and data was reviewed by the central pathology review board. This information was integrated into the second MDT meeting where data was discussed. Finally, histopathological analysis (in case surgical exploration was performed) or a 6-month follow-up period were used as the reference method.

## Outcomes

For analysis of the contribution of NGS to the diagnostic process only the patients with technically feasible NGS analysis were included. Diagnosis based on the information before and after integration of the NGS results was correlated with the final diagnosis. Differences were analyzed with a McNemar test. Proposed treatment plans before and after integration of NGS results were compared and classified as 'changes in treatment plan', 'confirmation of treatment plan' or 'no value of NGS results'.

All patients were included (including inconclusive NGS results) to determine the accuracy of 1) suggested diagnosis during first MDT meeting, 2) NGS results on itself, without interference of the tumor board and, 3) the suggested diagnosis during the second MDT meeting with the final diagnosis as a reference. Since *LGD* can be interpreted in several ways and does not provide a definitive diagnosis on itself, the accuracy is reported two-fold: all *LGD* incorrect, or all *LGD* correct.

## Power calculation

Assuming that NGS analysis has an additional diagnostic value of 20%, a total of 57 patients were required in the study (80% power and 2-sided significance level $\alpha = 0.05$).

## Results

In total, 57 patients were eligible for inclusion at the first MDT between 2015 and 2018. Four patients were excluded from further analysis due to incorrect inclusion: lymph node biopsy (n = 2), suspect GIST tumor (n = 1) and biopsy of suspect pancreatic liver metastasis (n = 1). Table 1 provides an overview of the patient characteristics.

NGS was technically feasible in 50 of the 53 patients, and further calculations were based on these 50 inclusions (94%). Fig 2A provides an overview of the pathogenic DNA variants

**Table 1. Patient characteristics.**

| | | N | % |
|---|---|---|---|
| **Total** | | **53** | |
| **Sex** | Female | 23 | 43 |
| | Male | 30 | 57 |
| **Age** | Mean, SD | 63.1 | 13.7 |
| **CA 19.9** | Median, range | 39 | 0–5126 |
| **Type of cytology material** | FNA | 40 | 76 |
| | Brush | 13 | 24 |
| **Cytology result** | Dysplasia | 10 | 19 |
| | Normal/atypia | 32 | 60 |
| | Inconclusive | 11 | 21 |
| **Reason NGS** | Cytology unsure | 31 | 59 |
| | MDT unsure | 22 | 41 |

PA: Pathology; MDT: Multidisciplinary team.

### NGS results and final diagnosis (n=50)

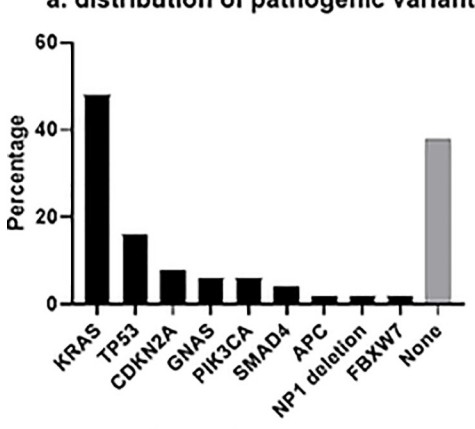

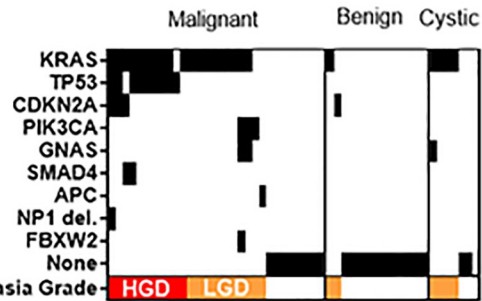

**Fig 2. Pathogenic variants identified in 50 patients.**

identified in these 50 patients. In 10 patients (20%), HGD was suspected after NGS analysis. All of which were classified as malignant tumors during histopathological analysis or follow-up. Two benign specimens showed LGD (KRAS mutation), both were classified as pancreatitis. Four out of six IPMNs showed LGD (Fig 2B and 2C). Two were classified as low grade IPMNs on pathology and the remaining two showed no worrisome features during the follow-up period.

Table 2 gives an overview of the probability diagnoses before and after integration of the NGS results, related to the final diagnosis. Integration of the NGS results significantly improved the probability diagnosis: from 16 incorrect diagnoses at the first MDT(32%) to three incorrect diagnoses after NGS integration (6%); (p-value < 0.001).

The treatment plan proposition changed in 16 cases (32%) after integration of the NGS results (Table 3). In eight patients, integration of the NGS results prevented surgical exploration (n = 6) and palliation (n = 2) as NGS did not show proliferative lesions (absent DNA variants) suggesting benign diseases. In all eight cases, follow-up confirmed the rightful change of treatment. Conversely, in one patient solely the NGS results concluded at least *HGD* and therefore the treatment plan was changed to surgical resection. In seven patients repeated EUS was

**Table 2. Diagnosis with and without NGS results correlated to final diagnosis.**

| | | Final diagnosis | | | |
|---|---|---|---|---|---|
| | | **Malignancy** | **Benign condition** | **Benign Cystic lesion** | **Accuracy** |
| **MDT 1: without NGS results** | **Malignancy** | **27** | 9 | 3 | |
| | **Benign condition** | 1 | **5** | 1 | |
| | **Benign Cystic lesion** | 2 | | **2** | 68% |
| **MDT 2: with NGS results** | **Malignancy** | **30** | 1 | 2 | |
| | **Benign condition** | | **13** | | |
| | **Benign Cystic lesion** | | | **4** | 94% |

MDT: Multidisciplinary team meeting.

Bold: Concordant results.

**Table 3. Changes in treatment plan after NGS integration.**

| | | Cross-over treatment | | | | | |
|---|---|---|---|---|---|---|---|
| | | Exploration | Neoadjuvant | Palliation | Follow-up | Repeat biopsy | Treatment changes |
| Without NGS results | Exploration | - | | | 6 | | 6 |
| | Neoadjuvant | | - | | | | 0 |
| | Palliation | | | - | 2 | | 2 |
| | Follow-up | 1 | | | - | | 1 |
| | Repeat biopsy | 1 | | 4 | 2 | - | 7 |
| | Total | | | | | | 16 |

prevented by integrating the NGS results, resulting in surgery (n = 1), palliation (n = 4) or follow-up (n = 2). In 12 patients (24%) the final diagnosis was altered after NGS integration.

During the second MDT meeting the NGS results were ignored in 12 patients (24%) (no incremental value NGS). Although no pathogenic variants were identified there was a strong clinical suspicion of a malignancy. In nine of these patients, NGS results were correctly ignored and final pathology concluded a malignancy. In the remaining three patients, no pathogenic variants were identified and final histologic assessment showed a pancreatitis (n = 1) and low grade IPMN (n = 2) (Table 4). In the remaining 22 patients (44%), the NGS results provided enough support to substantiate the initial clinical diagnosis, which was confirmed to be correct in all patients after follow-up or histopathology.

Results of the pathological variants identified in 50 patients. (a) shows the distribution of pathogenic variants, whereas (b) shows the pathogenic variants per patient, divided into three subgroups (malignant, benign and cystic lesions). The dysplasia grade is displayed in red (high grade dysplasia or HGD) and orange (low grade dysplasia, or LGD). (c) final diagnosis in these 50 patients after surgical exploration or 6-months follow-up.

## Discussion

In this study, the incremental value of NGS in patients with inconclusive diagnosis after regular diagnostic work-up (e.g. cytology, multi-phase CT) of a suspicious pancreatic lesion was assessed. NGS was feasible in 94% of patients. Integrating the NGS results in the diagnostic process increased the accuracy with 26% to a final, overall accuracy of 94%. Although this study was performed in a selected patient population where standard diagnostic work-up didn't suffice, adding NGS to this process was of incremental value in 76% of cases, either substantiating the diagnosis or changing the treatment. More importantly, no incorrect changes of treatment plan occurred indicating that integration of NGS is safe and feasible. Also, the

**Table 4. Treatment plan after NGS integration compared to final diagnosis.**

| | | Final diagnosis | | |
|---|---|---|---|---|
| | | Malignancy | Benign condition | Cystic lesion |
| With NGS results | Exploration | **13** | 1* | 4† |
| | Neoadjuvant | **5** | | |
| | Palliation | **10** | | |
| | FU | | **13** | 2 |
| | Repeat biopsy | 2 | | |

*Pancreatitis

†low grade IPMN, 2 were diagnosed as cystic lesion during the MDT meeting and too suspect, therefore, exploration/resection was chosen. The other two lesions were diagnosed as malignancies but turned out to be low grade IPMNs. Bold: Concordant.

study was performed in multiple centers which all have their own MDT, increasing data reproducibility.

In our previous study and other literature the accuracy of cytology is around 70% [7,8]. Moreover in this study's difficult-to-diagnose population the accuracy of cytology was only 17%, further supporting the possibility to integrate NGS in the diagnostic work-up for a more personalized management [9]. One major challenge in the diagnostic work-up of pancreatic lesions is the management of pancreatic cysts and IPMN in particular. Given the relatively limited number of patients with IPMNs in our study, conclusions regarding the added value of NGS in this cohort are not possible, but might be of value to the international discussion on the management of these lesions.

NGS has already gained an important role in cancer research and treatment and can aid in predicting response or resistance to cancer treatments. NGS has been studied in multiple cancer types. Predominantly in patients with advanced stage cancers and more specifically in lung cancer patients, given the multiple described pathways of oncogenic addiction. The technique aids in detection of mutations for applying the most suitable treatment [10]. Our previous study described a cross-section of 70 consecutive patients with suspect pancreatic masses discussed at the MDT meeting including those without much doubt at cytological assessment. We concluded that NGS showed a high accuracy with final pathology and that the addition of NGS changed the treatment plan in 10% of patient [5]. However, with rising expenditures on healthcare and an increasing complexity of oncologic care in particular, we felt that studying the patient population that likely benefits the most from this new diagnostic entity is of increasing importance. This follow-up study substantiates the hypotheses that NGS can be implemented as an addition to daily care for patient with suspect and difficult to diagnose pancreatic masses along with regular MDT meetings.

Most studies evaluating NGS determine the feasibility of NGS or provide insight in pathogenic variants for personalized systemic treatment. Studies focusing on the diagnostic yield of NGS in suspicious pancreatic lesions are scarce. Larson *et al*. evaluated different types of biopsy and reported on the success rate of NGS analysis of suspect pancreatic lesions or metastases [8]. Percutaneous biopsies of potential metastases were more adequate than FNA or fine needle biopsies of the suspect primary tumor. They performed NGS with the commercially available set FoundationOne (Foundation Medicine, Inc, Cambridge, Massachuttes).

NGS can often be performed if morphological assessment is not possible due to low cellular yield since only a limited quantity of material is necessary for NGS analysis. In this study this repeated EUS-FNA was spared in seven out of nine patients. Although the costs of NGS are relatively high, approximately €600 per sample, repetition of EUS-FNA in patients with inconclusive samples is more costly than performing NGS analysis. Suggesting cost-effectiveness of NGS in this patient population.

In our study, NGS was technically not feasible due to low cellular yield in 3/53 samples (6%), which was lower than the reported 29% (of the 76 samples) in the Larson study [8].

The results of NGS analysis could well be integrated into the diagnostic process and are an addition to the regular clinical and radiological data in this selected patient population. Only if *HGD* is suspected from NGS results (in this study in 21%), meaning the identification of class 4 or 5 *TP53* or *SMAD4* pathogenic variants, a malignancy is proven. In case of molecular *LGD*, or if no pathogenic variants are identified, the diagnosis and treatment plan are still challenging to establish. NGS analysis resulting in no pathogenic variants can be due to a truly benign condition. However, this can also be the result of a tumor without mutations present in the NGS panel or of sampling error (e.g. off-target FNA, small tumor size, extensive inflammatory component) [5]. Secondly, in case of *LGD*, the identification of a sole *KRAS* pathogenic variant can indicate a pancreatitis, IPMN or malignancy.

Although not used in this study, NGS can also be applied on cystic fluid [11]. Rosenbaum *et al*. analyzed fluid from 113 cysts, where 67 cysts showed pathogenic variants [12]. A significant difference in identification of *KRAS* pathogenic variants was present between non-mucinous cysts (12.5% of samples) and mucinous cysts/IPMN (100% of samples). Furthermore, in 33% of the IPMN samples a *GNAS* pathogenic variant was also identified. In the IPMNs with a malignant component additional pathogenic variants in the following genes were identified: *TP53*, *CDKN2A*, *SMAD4*, *NOTCH1*. Singi *et al*. analyzed fluid from 308 cysts and also identified *KRAS* and *GNAS* pathogenic variants in IPMNs and MCNs and showed *TP53*, *PIK3CA* and *PTEN* in neoplasia [11]. Volckmar *et al*. published the first results of the ZYSTEUS study that investigates whether detection of driver mutations in IPMN patients by liquid biopsy (cystic fluid) is technically feasible [13]. Comparable results were obtained, in 12 IPMNs *KRAS* (n = 12) and *GNAS* (n = 4) pathogenic variants were identified. In three pseudocysts no pathogenic variants were identified. Lastly, Suenaga *et al*. studied whether patients under surveillance for IPMNs would benefit from pancreatic juice punction for NGS analysis [14]. In case a *TP53* or *SMAD4* pathogenic variant was identified the lesion was resected. The results of all these studies are in line with ours; the identification of a *SMAD4* or *TP53* is suspect for a malignancy, identification of a *CDKN2A*, *NOTCH1*, *PIK3CA* or *PTEN* is also suggestive for a malignancy. Based on our results and the abovementioned studies the NGS results of patients with a pathogenic variant in *PIK3CA* could also be classified as *HGD*. In our cohort, 3 patients carried a PIK3CA mutation and were later diagnosed as a malignancy. A pathogenic variation in *GNAS* is suggestive for an IPMN and the interpretation of a pathogenic variant in *KRAS* remains somewhat challenging since this can be seen in in IPMNs, malignancies but also in pancreatitis.

In conclusion, NGS can help to substantiate the adequate diagnosis and determine the suitable treatment for suspect pancreatic lesions in addition to standard pathology or imaging assessment in patients with inconclusive cytopathology. It could therefore provide a more personalized approach to the diagnostic work-up. By integrating NGS in the diagnostic process, an accuracy of 94% for correct diagnosing was reached. NGS is of incremental value by providing molecular confirmation of neoplasia for preoperative chemotherapy or surgery, while preventing repeat EUS-FNA/brushing or futile surgery in patients with benign conditions.

## Supporting information

**S1 File.**
(PDF)

## Author Contributions

**Conceptualization:** Babs G. Sibinga Mulder, Quisette P. Janssen, Bas Groot Koerkamp, Rutger Quispel, Bert A. Bonsing, Alexander L. Vahrmeijer, Casper H. J. van Eijck, Daphne Roos, Lars E. Perk, Erwin van der Harst, Peter-Paul L. O. Coene, Michail Doukas, Marie-Louise F. van Velthuysen, Arantza Farina Sarasqueta, J. Sven D. Mieog.

**Data curation:** Friso B. Achterberg, Babs G. Sibinga Mulder, Quisette P. Janssen, Lieke Hol, Rutger Quispel, Bert A. Bonsing, Alexander L. Vahrmeijer, Lars E. Perk, Erwin van der Harst, Peter-Paul L. O. Coene, Michail Doukas, Arantza Farina Sarasqueta, Hans Morreau, J. Sven D. Mieog.

**Formal analysis:** Babs G. Sibinga Mulder, Quisette P. Janssen, Lieke Hol, J. Sven D. Mieog.

**Funding acquisition:** Babs G. Sibinga Mulder, Alexander L. Vahrmeijer, J. Sven D. Mieog.

**Investigation:** Babs G. Sibinga Mulder, Quisette P. Janssen, Alexander L. Vahrmeijer, Lars E. Perk, Erwin van der Harst, Peter-Paul L. O. Coene, Michail Doukas, Frank M. M. Smedts, Mike Kliffen, Valeska Terpstra, Arantza Farina Sarasqueta, Hans Morreau, J. Sven D. Mieog.

**Methodology:** Babs G. Sibinga Mulder, Alexander L. Vahrmeijer, Casper H. J. van Eijck, Daphne Roos, Lars E. Perk, Erwin van der Harst, Peter-Paul L. O. Coene, Michail Doukas, Frank M. M. Smedts, Mike Kliffen, Valeska Terpstra, Arantza Farina Sarasqueta, Hans Morreau, J. Sven D. Mieog.

**Project administration:** Friso B. Achterberg, Babs G. Sibinga Mulder, Quisette P. Janssen, Lieke Hol, Frank M. M. Smedts, Marie-Louise F. van Velthuysen, Valeska Terpstra.

**Resources:** Babs G. Sibinga Mulder, Bas Groot Koerkamp, Lars E. Perk, J. Sven D. Mieog.

**Software:** Friso B. Achterberg, Babs G. Sibinga Mulder.

**Supervision:** Bas Groot Koerkamp, Rutger Quispel, Bert A. Bonsing, Alexander L. Vahrmeijer, Casper H. J. van Eijck, Daphne Roos, Lars E. Perk, Peter-Paul L. O. Coene, Marie-Louise F. van Velthuysen, Valeska Terpstra, Arantza Farina Sarasqueta, Hans Morreau, J. Sven D. Mieog.

**Validation:** Friso B. Achterberg, Babs G. Sibinga Mulder, Michail Doukas, Frank M. M. Smedts, Mike Kliffen, Marie-Louise F. van Velthuysen, Valeska Terpstra, Arantza Farina Sarasqueta, Hans Morreau, J. Sven D. Mieog.

**Visualization:** Friso B. Achterberg, Babs G. Sibinga Mulder, Alexander L. Vahrmeijer, Frank M. M. Smedts, Valeska Terpstra, Arantza Farina Sarasqueta, J. Sven D. Mieog.

**Writing – original draft:** Friso B. Achterberg, Babs G. Sibinga Mulder, Quisette P. Janssen, Bas Groot Koerkamp, Lieke Hol, Rutger Quispel, Bert A. Bonsing, Alexander L. Vahrmeijer, Casper H. J. van Eijck, Daphne Roos, Lars E. Perk, Erwin van der Harst, Peter-Paul L. O. Coene, Michail Doukas, Frank M. M. Smedts, Mike Kliffen, Marie-Louise F. van Velthuysen, Valeska Terpstra, Arantza Farina Sarasqueta, Hans Morreau, J. Sven D. Mieog.

**Writing – review & editing:** Friso B. Achterberg, Bas Groot Koerkamp, Rutger Quispel, Bert A. Bonsing, Alexander L. Vahrmeijer, Casper H. J. van Eijck, Peter-Paul L. O. Coene, J. Sven D. Mieog.

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
