## [Decision Letter · Decision Letter 0]

28 Sep 2022

PONE-D-22-23812Targeted Next-Generation Sequencing has incremental value in the diagnostic work-up of patients with suspect pancreatic masses; a multi-center prospective cross sectional studyPLOS ONE

Dear Dr.Mieog ,

Thank you for submitting your manuscript to PLOS ONE. After careful consideration, we feel that it has merit but does not fully meet PLOS ONE’s publication criteria as it currently stands. Therefore, we invite you to submit a revised version of the manuscript that addresses the points raised during the review process.

We look forward to receiving your revised manuscript.

Kind regards,

Fabrizio D'Acapito, Ph.D,M.D.

Academic Editor

PLOS ONE

Journal Requirements:

"JSDM: Dutch Cancer Society (grant UL2015-7665)"

"Dutch Cancer Society (grant UL2015-7665)"

"JSDM: Dutch Cancer Society (grant UL2015-7665)"

"No"

7. We note that you have included the phrase “data not shown” in your manuscript. Unfortunately, this does not meet our data sharing requirements. PLOS does not permit references to inaccessible data. We require that authors provide all relevant data within the paper, Supporting Information files, or in an acceptable, public repository. Please add a citation to support this phrase or upload the data that corresponds with these findings to a stable repository (such as Figshare or Dryad) and provide and URLs, DOIs, or accession numbers that may be used to access these data. Or, if the data are not a core part of the research being presented in your study, we ask that you remove the phrase that refers to these data.

Additional Editor Comments:

I believe the paper is well written and clearly reports the method and results.

The proposed method could become part of the tools available to multidisciplinary teams that face daily decision-making difficulties related to pancreatic neoplasms. The advantage also seems clear in terms of cost.

I believe that by making some additions as suggested by reviewer 2 (particularly related to the weight and importance that PMID 28775172 had on the development of this work) the paper can be accepted

Reviewers' comments:

Reviewer's Responses to Questions

**Comments to the Author**

1. Is the manuscript technically sound, and do the data support the conclusions?

Reviewer #1: Yes

Reviewer #2: Partly

2. Has the statistical analysis been performed appropriately and rigorously? 

Reviewer #1: Yes

Reviewer #2: Yes

3. Have the authors made all data underlying the findings in their manuscript fully available?

Reviewer #1: Yes

Reviewer #2: Yes

4. Is the manuscript presented in an intelligible fashion and written in standard English?

Reviewer #1: Yes

Reviewer #2: Yes

5. Review Comments to the Author

Reviewer #1: The paper is well written and clear. I have no major nor minor observations. The topic is interesting and probably this kind of research is going to add something in the day-by-day pancreatology practice.

Reviewer #2: The authors present the outcomes of a multicenter, prospective study aiming to propsepdetermine the incremental diagnostic value of NGS

in the diagnostic process and treatment plan proposition in patients with a suspect pancreatic lesion with an inconclusive cytopathologic diagnosis. Based on their outcomes they conclude that NGS can help to substantiate the adequate diagnosis and determine the suitable treatment for suspect pancreatic lesions in addition to standard pathology or imaging assessment in

patients with inconclusive cytopathology.

This is quite an interesting study, which seems well-stuctured and written.

- One major issue I advise the authors to correct is to add a distinct strengths and most importantly limitations section.

- More information on the PMID 28775172 should be provided to the readership. What is the added value of this study?

- Equally reference [5] needs to be corrected.

6. PLOS authors have the option to publish the peer review history of their article (what does this mean?). If published, this will include your full peer review and any attached files.

Reviewer #1: No

Reviewer #2: No

---

## [Author Response · Author response to Decision Letter 0]

24 Dec 2022

Dear reviewers, 

We above all thank you for reviewing our manuscript and therefore contributing to the quality of our manuscript. Please find below a detailed description and our response to the comments provided by the reviewers 

1. Response to reviewer #2 and the editor concerning our previous study (PMID 28775172)

Answer: Thank you for this comment. We feel that this comment in particular is of added value to the manuscript. We therefore have added a new paragraph to the discussion section. 

‘Our previous study described a cross-section of 70 consecutive patients with suspect pancreatic masses discussed at the MDT meeting including those without much doubt at cytological assessment. We concluded that NGS showed a high accuracy with final pathology and that the addition of NGS changed the treatment plan in 10% of patients. However, with rising expenditures on healthcare and an increasing complexity of oncologic care in particular, we felt that studying the patient population that likely benefits the most from this new diagnostic entity is of increasing importance. This follow-up study substantiates the hypotheses that NGS can be implemented as an addition to daily care for patient with suspect and difficult to diagnose pancreatic masses along with regular MDT meetings.’

2. Response to reviewer #2 concerning a distinct section where the strengths and limitations are discussed. 

Answer: We concur that it is very important to discuss the strengths and limitations of a study in the discussion section. We are of the opinion that our study’s main strength is also its limitation; the selected patient population. Although our approach might be applicable to other patient populations with difficult to diagnose tumors which are discussed in MDTs, we are unaware of the possibilities that NGS might provide in these other diseases. 

We have added the following two sections to the manuscript: 

‘Also, the study was performed in multiple centers which all have their own MDT, increasing data reproducibility.’

‘Although our study is comprised of a limited number of patients and mainly focused on differentiating between malignant tumors and benign disease, our approach might be well applicable to other diseases. One major challenge in the diagnostic work-up of pancreatic lesions is the management of pancreatic cysts and IPMN in particular. Given the relatively limited number of patient with IPMNs in our study, conclusions regarding the added value of NGS in this cohort are not possible, but might be of value to the international discussion on the management of these lesions.’

---

## [Decision Letter · Decision Letter 1]

12 Jan 2023

Targeted Next-Generation Sequencing has incremental value in the diagnostic work-up of patients with suspect pancreatic masses; a multi-center prospective cross sectional study

PONE-D-22-23812R1

Dear Dr. Mieog,

We’re pleased to inform you that your manuscript has been judged scientifically suitable for publication and will be formally accepted for publication once it meets all outstanding technical requirements.

Kind regards,

Fabrizio D'Acapito, Ph.D,M.D.

Academic Editor

PLOS ONE

Additional Editor Comments (optional):

Congratulations on the paper. I think it can really help in the practice of multidisciplinary teams dealing with pancreatic disease.

Reviewers' comments:

Reviewer's Responses to Questions

**Comments to the Author**

1. If the authors have adequately addressed your comments raised in a previous round of review and you feel that this manuscript is now acceptable for publication, you may indicate that here to bypass the “Comments to the Author” section, enter your conflict of interest statement in the “Confidential to Editor” section, and submit your "Accept" recommendation.

Reviewer #1: All comments have been addressed

Reviewer #2: All comments have been addressed

2. Is the manuscript technically sound, and do the data support the conclusions?

Reviewer #1: Yes

Reviewer #2: Yes

3. Has the statistical analysis been performed appropriately and rigorously? 

Reviewer #1: Yes

Reviewer #2: Yes

4. Have the authors made all data underlying the findings in their manuscript fully available?

Reviewer #1: Yes

Reviewer #2: Yes

5. Is the manuscript presented in an intelligible fashion and written in standard English?

Reviewer #1: Yes

Reviewer #2: Yes

6. Review Comments to the Author

Reviewer #1: (No Response)

Reviewer #2: The authors have adequately adressed my remarks in their revised manuscript. This work is now suitable for publication.

7. PLOS authors have the option to publish the peer review history of their article (what does this mean?). If published, this will include your full peer review and any attached files.

Reviewer #1: No

Reviewer #2: No

---

## [Editor Report · Acceptance letter]

17 Jan 2023

PONE-D-22-23812R1 

Targeted Next-Generation Sequencing has incremental value in the diagnostic work-up of patients with suspect pancreatic masses; a multi-center prospective cross sectional study 

Dear Dr. Mieog:

I'm pleased to inform you that your manuscript has been deemed suitable for publication in PLOS ONE. Congratulations! Your manuscript is now with our production department. 

Kind regards, 

on behalf of

Dr. Fabrizio D'Acapito 

Academic Editor

PLOS ONE